# Disease transmission and introgression can explain the long-lasting contact zone of modern humans and Neanderthals

Gili Greenbaum [1]*, Wayne M. Getz [2,3], Noah A. Rosenberg[1], Marcus W. Feldman[1], Erella Hovers [4,5] & Oren Kolodny [1,6]*

Neanderthals and modern humans both occupied the Levant for tens of thousands of years prior to the spread of modern humans into the rest of Eurasia and their replacement of the Neanderthals. That the inter-species boundary remained geographically localized for so long is a puzzle, particularly in light of the rapidity of its subsequent movement. Here, we propose that infectious-disease dynamics can explain the localization and persistence of the inter-species boundary. We further propose, and support with dynamical-systems models, that introgression-based transmission of alleles related to the immune system would have gradually diminished this barrier to pervasive inter-species interaction, leading to the eventual release of the inter-species boundary from its geographic localization. Asymmetries between the species in the characteristics of their associated 'pathogen packages' could have generated feedback that allowed modern humans to overcome disease burden earlier than Neanderthals, giving them an advantage in their subsequent spread into Eurasia.

[1] Department of Biology, Stanford University, Stanford, CA 94305, USA. [2] Department of Environmental Science, Policy, and Management, University of California, Berkeley, CA 94720, USA. [3] School of Mathematical Sciences, University of KwaZulu-Natal, Durban 4041, South Africa. [4] The Institute of Archaeology, The Hebrew University of Jerusalem, 9190501 Jerusalem, Israel. [5] Institute of Human Origins, Arizona State University, Tempe, AZ 85287, USA. [6] Department of Ecology, Evolution, and Behavior, The Hebrew University of Jerusalem, 9190401 Jerusalem, Israel. *email: gilig@stanford.edu; oren.kolodny@mail.huji.ac.il

The lineages leading to modern humans (*Homo sapiens*; henceforth 'Moderns') and Neanderthals (*H. neanderthalensis*) diverged 500–800 kya, with Neanderthals inhabiting Eurasia and Moderns inhabiting Africa[1–3]. Migrating out of Africa, Moderns reached the Levant tens of thousands of years prior to their further spread throughout Eurasia, whereas Neanderthals seem never to have spread south of the Levant[4–6]. The two species most likely interacted in this region, at least intermittently, for extended periods of time, along a fairly narrow front[4,7–9]. This contact is generally believed to have been accompanied by a low level of repeated interbreeding[8–12]. Later, around 45–50 kya, Moderns spread further into Eurasia, replacing the Neanderthals within a few thousand years, by about 39 kya[5,7,13,14] (Fig. 1).

The replacement of Neanderthals by Moderns has been extensively studied and debated (e.g. refs. [15–18]). Less attention, however, has been given to the fact that contact in the Levant was made much earlier than the initiation of the replacement phase[4,7]. Therefore, an important question remains open: irrespective of the driving mechanisms of the eventual replacement, what caused the apparent delay in the commencement of the replacement process? This question is especially puzzling because the time frame during which the interaction front was geographically confined to the Levant[4,7,19] was much longer than the time frame during which replacement occurred across Eurasia.

In this study, we propose that infectious disease dynamics provide a possible solution to this puzzle. We suggest that, following contact, transmissible diseases might have played a prominent role in stabilizing and localizing the inter-species population dynamics in the Levant. We also propose that these dynamics might have resulted in transmissible diseases playing an important role during the later replacement phase. We propose that after several hundred thousand years of largely independent evolution, Neanderthals and Moderns likely acquired immunity and tolerance to different suites of pathogens—a temperate pathogen package in the case of Neanderthals and a tropical pathogen package in Moderns. The re-establishment of contact in the Levant would have resulted in exposure of each species to novel pathogens carried by the other species. In turn, these pathogens could have spread to the new susceptible hosts, placing a considerable disease burden on both species.

Whether the two groups constitute two species or two subspecies is a matter of debate[20]; for our purposes, all that matters is that they were geographically distinct for a long period of time, and we refer to them as species for convenience. Nevertheless, many genomic studies of Neanderthals and Moderns have detected a signature of introgression (gene flow) between the species, to the extent that Neanderthal sequences may represent 1–3% of present-day non-African modern human genomes[11,21,22]. That inter-species contact was potentially sufficient to allow for gene flow suggests that disease transmission between the species was likely. Although many of the pathogens that may have been transmitted may not exist today, several genomic studies record potential signatures of events in which pathogens were transmitted between Moderns and Neanderthals, or between Moderns and other archaic humans[23–27]. Moreover, studies have identified signatures of positive selection on putatively introgressed Neanderthal genes in Moderns, particularly in genomic regions such as the MHC complex that are associated with the immune system[2,22,26,28–32]. These findings suggest that disease burden due to inter-species pathogen transmission was significant.

We use mathematical models of inter-species interaction to explore the possible ecological and demographic consequences of exposure to novel pathogen packages upon inter-species contact, and the coupling of this exposure with immune-related adaptive introgression. Due to the disease burden imposed by contact with the other species, bands of individuals in each species would have been strongly disadvantaged when migrating into regions dominated by the other one, because such migrations would have resulted in increased exposure to novel pathogens. Additionally, disease burden—realized as recurring epidemics, greater endemic pathogen load, or both—would have decreased population densities of both species near the contact front, further reducing the likelihood and motivation for bands of one species to migrate into the range of the other species. The interaction between the two species would thus have been limited to a circumscribed region, which would have been geographically localized by the disease dynamics (Fig. 1). This front may have been constant[4,33] or intermittent[7,34,35], alternately strenghening and weakening in time and space during the period of contact—in the Levant and perhaps also in the Arabian peninsula[36]. Some contact between the two species would have continued along this front, resulting in pathogen spillovers between the species, but also in occasional interbreeding through which transmission of immune-related genes would have occurred. Under these circumstances, bands of each species close to the interaction front would gradually have been able to adapt and acquire tolerance to the novel pathogens, through adaptive introgression as well as by selection on de novo immune-related mutations. Eventually, this process would have reduced disease burden, diminishing the effect of disease transmission dynamics and allowing other processes to drive population dynamics. At this point, the barrier to full inter-species contact and cross-regional migration would have been removed, destabilizing the front of interaction and enabling the species dynamics that eventually led to Neanderthal replacement. Disease dynamics, as we will show, are sufficient to explain the extended existence of a stationary interaction front in the Levant, although our analysis does not preclude the importance of additional processes, such as adaptation of the two species to their respective local environments and inter-species competition (e.g. refs. [37–39]; see the "Discussion" section).

Once the interaction front was destabilized, presumably around 45–50 kya, other processes, previously overshadowed by disease burden, would then have been responsible for the replacement of Neanderthals by Moderns[17,18,40], although it has been suggested that disease transmission dynamics could also have played a prominent role in the replacement process[31,41,42]. Our model supports this possibility. Conditions relating to disease dynamics need not have been symmetric between Moderns and Neanderthals—for example, the Moderns' tropical pathogen package may have been more burdensome to the Neanderthals than the Neanderthals' temperate pathogen package was to Moderns, following the pattern of decreasing pathogen burden with latitude[43,44]. Hence, Moderns may have overcome the disease burden from contact sooner than Neanderthals. This asymmetry would have eventually allowed bands of Moderns to migrate into the Neanderthal regions unhindered by novel transmissible diseases, while carrying contagious diseases to which the Neanderthals were not yet immune. Moreover, after the historical front of interaction was crossed and migration reached deeper into Eurasia, this relative Modern advantage would have increased further, as Neanderthal bands encountered far from the initial contact zone would have been intolerant to the entirety of the novel pathogen package spread by the Moderns. We thus suggest, following patterns that occurred multiple times in the colonial era when two long-separated populations renewed contact[27,45–48], that the replacement of Neanderthals by Moderns may have been facilitated by pathogens to which Moderns were largely immune but to which the Neanderthals were vulnerable.

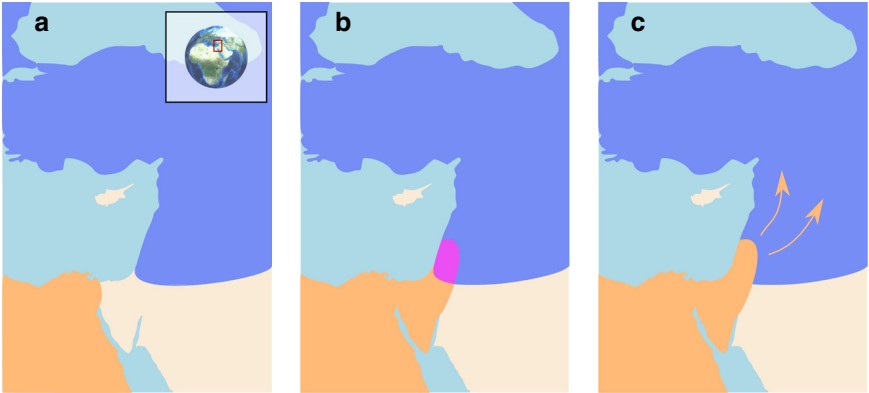

**Fig. 1** Schematic representation of interaction between modern humans and Neanderthals. The geographic range of modern humans appears in orange, that of Neanderthals is in blue, and the contact zone in the Levant is in purple. **a** Neanderthals and Moderns were separated for several hundred thousand years, with Moderns co-evolving with tropical pathogens in Africa and Neanderthals co-evolving with temperate pathogens in Eurasia. **b** As early as 180–120 kya, Moderns began migrating out of Africa into the Levant[4-6]. Their range remained restricted to this region for tens of thousands of years, during which they interacted intermittently with Neanderthals[4,7-9]. We propose that during this period, each species was exposed to novel pathogen packages carried by the other species and experienced disease burden. Note that the contact zone depicted here may have been larger, possibly including regions in the Arabian Peninsula[36]. **c** Around 45–50 kya, the inter-species dynamics destabilized and Moderns began expanding further into Eurasia. Within several thousand years, Moderns replaced Neanderthals throughout Eurasia[4,5,7,13,14]

## Results and discussion

**Modeling disease transmission and introgression dynamics.** Our main proposition is that a persistent Modern–Neanderthal front of interaction in the Levant can be explained by disease burden that prevented each species from expanding into the region dominated by the other. We propose that this front could eventually have collapsed due to immune-related adaptive introgression. In order to (i) demonstrate the feasibility of this scenario, (ii) understand the consequences of variation in the details of this process, (iii) investigate the impact of feedback between disease and gene transmission, and (iv) explore the robustness of our proposition, we model disease transmission and introgression dynamics between two species using dynamical-systems models.

We first explore a model that bridges two independently treated time-scales, ecological and evolutionary ('two-time-scales model'), as summarized in Fig. 2. At the ecological time scale, disease spillovers of novel pathogens, whose impact is measured as the proportion of the non-tolerant population that is infected by each epidemic, are modeled in terms of between-species and within-species contact rates (denoted $\beta_{ij}$, the contact rates from species $i$ to species $j$), using a well-mixed SIR-modeling framework (Eqs. (3)–(8)). On the evolutionary time scale, the two species respond to disease burden ($D_i$ for species $i$), which we measure as proportional both to the number (or diversity) of novel pathogens to which the species is exposed ($P_i$) and to the impact of a pathogen at the ecological time scale ($F_i$); Eqs. (1) and (2). At each time step in the evolutionary time scale, this response is modeled as an adjustment of the contact rates ($\beta_{ij}$), in proportion to the disease-burden experienced (Eqs. (10)–(13)); for example, in response to the impact of disease that they experience, groups may adjust their tendency to accept individuals from other groups from the same or from different species[49]. Additionally, also at the evolutionary time-scale, through inter-species contact, immune-related genes are exchanged, decreasing the potential impact of transmissible diseases over time. We assume that the rate of this adaptive introgression is proportional to the inter-species contact rates, and over time reduces the number of novel pathogens $P_i$ to which each species is vulnerable (Eqs. (14) and (15)).

**Symmetric conditions at the time of initial contact.** We first characterize the general behavior of our model by exploring a scenario in which initial conditions are symmetric for the two species. An outcome of this scenario is presented in Fig. 3. A more thorough exploration of the parameter space, with different parameterizations of the model, appears in Supplementary Note 1, with qualitatively similar results.

When the two species first come into contact, high contact rates and vulnerability to many novel pathogens result in high disease burden (Eqs. (1) and (2)), which elicits a rapid response of large effect, and the species lower both within-species and between-species contact rates (Fig. 3, orange phase). Following this initial response, the species maintain stable but low contact rates for an extended period (Fig. 3a, green phase). During this period, the pathogen package $P_i$ is reduced by adaptive introgression (Fig. 3b, green phase; Eqs. (14) and (15)), but disease burden is kept low and close to constant by the continuous minor adjustment of contact rates ($\beta_{ij}$) made by the species (Fig. 3c, green phase; Eqs. (10)–(13)). We interpret the disease burden at this phase of the dynamics as limiting each species from expanding into regions dominated by the other, thereby leading to geographic localization of the interaction front. This localization is further reinforced by the tendency of bands to isolate themselves, as demonstrated by low contact rates (Fig. 3a, green phase).

Eventually, disease burden is removed due to introgression ($D_i(t) \rightarrow 0$ in Eqs. (1) and (2), because $P_i(t) \rightarrow 0$ in Eqs. (14) and (15)). Once disease burden is removed, the dynamics destabilize, and the species, released from disease burden, begin to recover and return to the initial state that existed before contact was made (Fig. 3, blue phase). This destabilization would then allow other dynamics, previously overshadowed by disease burden, to play out.

**Asymmetric conditions at the time of initial contact.** It is unlikely that conditions were symmetric at the time that Moderns and Neanderthals came into contact in the Levant, for example with respect to different pathogen co-evolution trajectories in Africa and in Eurasia. We therefore model asymmetric initial conditions, focusing for tractability on one possible aspect of asymmetry at a time. Fig. 4 explores the effect of different initial

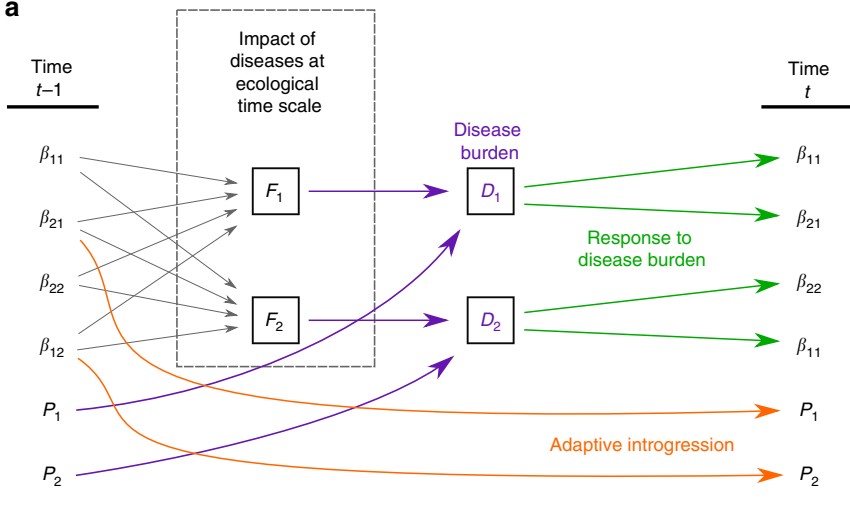

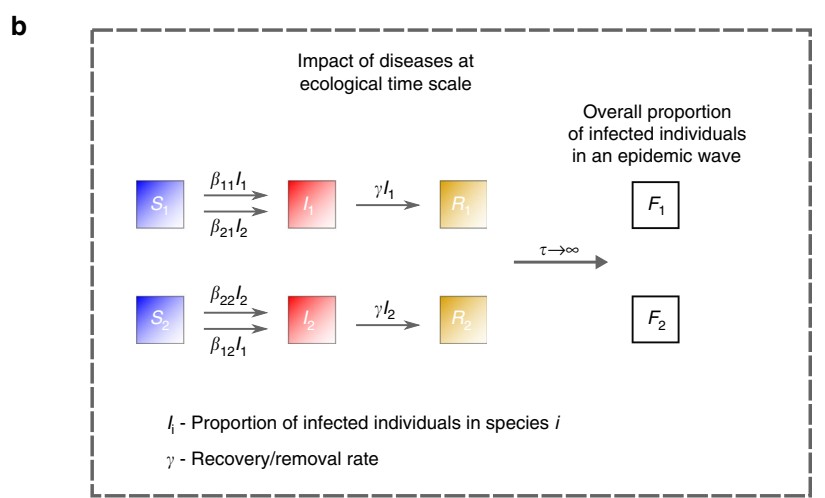

**Fig. 2** Schematic representation of the two-time-scales model. The model describes disease and introgression dynamics between two species. The figure shows the transition between two time steps at the evolutionary time scale, $t-1$ and $t$. **a** Transmission rates $\beta_{ij}$ ($i = j$, within-species transmission; $i \neq j$, between-species transmission from species $i$ to species $j$) determine the average impact of pathogens $F_i$ at the ecological time scale, according to Eqs. (3)-(9). The pathogen package size $P_i$ and the average impact of each pathogen $F_i$ determine the disease burden $D_i$ (purple), according to Eqs. (1) and (2). The species respond to disease burden by adjusting contact rates $\beta_{ij}$ (green), according to Eqs. (10)–(13). Inter-species contact results in gene flow and adaptive introgression, reducing pathogen package sizes $P_i$ (orange), according to Eqs. (14) and (15). **b** Impact at the ecological time scale (dashed box in **a**) is modeled in an SIR epidemic framework, as the average impact of an epidemic. Individuals transition from state S (susceptible) to state I (infectious) from either within-species infections or between-species infections (Eqs. (3)–(8)). Individuals in state I transition to state R (recovered/removed) at rate $\gamma$ (Eqs. (3)–(8)). The impact of an epidemic $F_i$ is measured as the overall proportion of individuals in species $i$ infected throughout the run of the epidemic (9)

pathogen package sizes ($P_2(0) > P_1(0)$), and asymmetry in initial contact rates is explored in Supplementary Note 1.

When $P_2(0) > P_1(0)$, species 2 experiences higher disease burden at the time of contact ($D_2(0) > D_1(0)$) due to the higher number of novel pathogens to which it is exposed (Eqs. (1) and (2)). This initial condition is reflected in a stronger initial response to disease burden, and lower within-species and incoming between-species contact rates (Fig. 4). The dynamics then enter a stable phase, similar to that in the symmetric case, but with some differences. First, species 1 retains slightly higher contact rates than species 2 during the stable phase (Fig. 4a), because the disease burden species 1 experiences is lower than that of species 2. Second, due to these higher contact rates, and specifically the between-species

incoming contact rates, the rate of adaptive introgression into species 1 is higher than into species 2 ($c\beta_{21}(t) > c\beta_{12}(t)$ during this period; Eqs. (14) and (15)). This pattern, in conjunction with the initial difference in pathogen package sizes, permits species 1 to overcome disease burden, reaching $P_1 = 0$, earlier than species 2 reaches $P_2 = 0$. Different parameterizations of the model yield similar qualitative results (Supplementary Note 1).

These dynamics, which are qualitatively similar to Figure 4 with other sources of asymmetries (Supplementary Note 1), mean that the species that was initially less vulnerable, species 1, overcomes disease burden sooner, and is therefore released sooner from the disease limitation enabling expansion into regions dominated by species 2. At the time that species 1 is

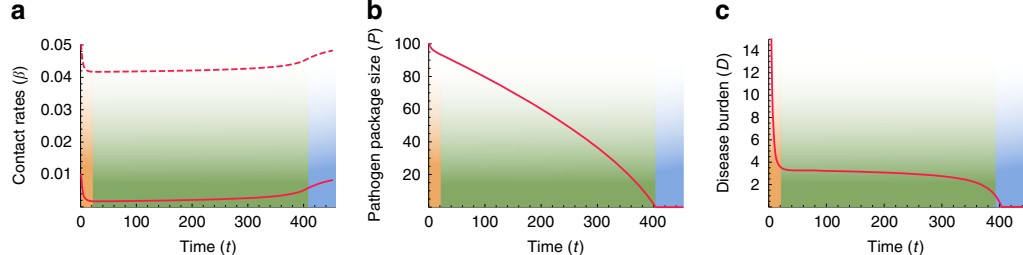

**Fig. 3** Disease-introgression dynamics between two species with symmetric initial conditions. The dynamics are described by the two-time-scales model (Fig. 2), following Eqs. (1)–(15). The initial conditions are assumed to be symmetric for the two species ($\beta_{ii} = \beta_{jj}$ and $\beta_{ij} = \beta_{ji}$), and therefore, the dynamics for the two species are identical, indicated by red curves. **a** Between-species contact rates ($\beta_{ij}$, $i \neq j$) appear in dashed curves, and within-species contact rates ($\beta_{ii}$) appear in dashed curves (Eqs. (10)–(13)). **b** Pathogen package size ($P_i$), measured as number or diversity of novel pathogens to which the species are vulnerable (Eqs. (14) and (15)). **c** Disease burden ($D_i$), proportional to pathogen package size $P_i$ and to the impact of diseases on each of the species, $F_i$ (Eqs. (1) and (2)). Three phases are observed in the dynamics: (1) Initial response to heavy disease burden, with decreased contact rates (orange); (2) Long-lasting stable phase with low but steady levels of disease burden (green); (3) Destabilization following the release from disease burden and recovery to initial conditions (blue). The parameters for the scenario modeled are $\beta_{11}(0) = \beta_{22}(0) = 0.05$; $\beta_{12}(0) = \beta_{21}(0) = 0.01$; $\gamma = 0.045$; $P_1(0) = P_2(0) = 100$, with scaling parameters $a = 5 \times 10^{-5}$, $b = 0.02$, and $c = 100$

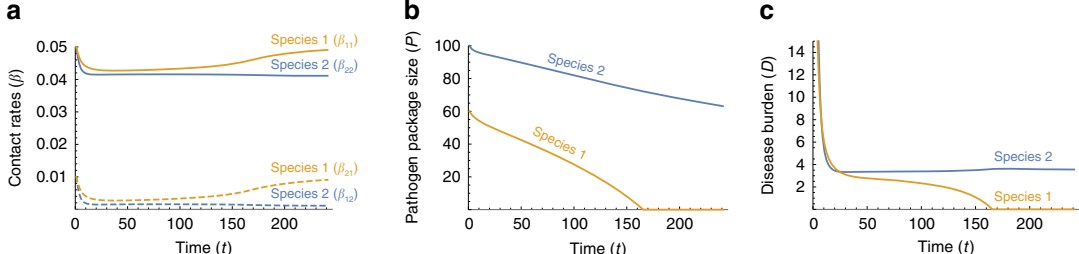

**Fig. 4** Disease-introgression dynamics between two species with asymmetric initial conditions. The dynamics of the two-time-scales model are described by Eqs. (1)–(15), with parameters identical to those in Fig. 3 except that species 1 (orange) is initially exposed to a smaller pathogen package than species 2 (blue), i.e. species 2 experiences more novel pathogens at the time of contact; $P_1(0) = 60$, $P_2(0) = 100$. **a** Between-species contact rates ($\beta_{ij}$, $i \neq j$) appear as dashed curves, and within-species contact rates ($\beta_{ii}$) appear as continuous curves (Eqs. (10)–(13)). **b** Pathogen package size, $P_i$. **c** Disease burden, $D_i$. Species 1 experiences a lower disease burden at the initial phase, and therefore responds less strongly and maintains higher contact rates (in **a**). This higher contact rate leads to faster rates of adaptive introgression into species 1 (in **b**). Finally, species 1 is released from disease burden sooner than species 2 (in **c**), while still carrying many pathogens to which species 2 is vulnerable (in **b**)

released from the novel disease burden, species 2 is still vulnerable to many novel pathogens carried by species 1 (Fig. 4b).

**Feedback between disease transmission and introgression.** Disease spread between species, in general, depends on the within-species and between-species contact rates, with higher contact rates resulting in greater disease impact. The level of contact between species, and specifically the amount of inter-breeding, also determines the expected rate of gene flow, and higher contact rates are expected to result in more rapid adaptive introgression. That both of these phenomena—disease transmission and adaptive introgression—depend on inter-species contact rates, but have opposite effects on the contact rates, generates feedback that complicates prediction of disease and introgression dynamics (Fig. 2).

In our model, high disease burden for a species negatively influences contact rates ($-a_{ij}D_j$ in Eqs. (10)–(13)), and lower between-species contact rates in turn negatively influence the rate of adaptive introgression for that species ($-c\beta_{ij}$ in Eqs. (14) and (15)). Therefore, the model generates a positive feedback that amplifies any initial differences in disease burden over time. In the case of asymmetric pathogen package sizes, the effect of differences in disease burden on the rate of adaptive introgression can be seen in the differing slopes of the pathogen package size trajectories of species 1 and 2 (Fig. 4b). Species 1, which at the

time of contact experiences a lower disease burden, also adapts faster to the novel pathogens due to its maintenance of higher incoming between-species contact rates ($\beta_{21} > \beta_{12}$; Fig. 4a). In the scenario shown, this asymmetry results in the initial difference of $P_2(0) - P_1(0) = 40$ pathogens in the pathogen package sizes at the time of contact (the units could be different, such as pathogen diversity, but we focus on the relative outcomes without attention to the interpretation of the units), growing to about $P_2(t) - P_1(t) = 75$ pathogens by the time species 1 has overcome disease burden (Fig. 4b).

Another implication of this feedback is observed when species 1 is released from disease burden, i.e. when $P_1 = 0$ is reached (Fig. 4b). At this point, species 1 begins to increase its contact rates (Fig. 4a), and in so doing increases the impact of disease in the two-species system (Eqs. (3)–(9)). However, at this point, only species 2 is vulnerable to the novel disease, and therefore, only species 2 is forced to respond by further reducing contact rates (Fig. 4c), further slowing the rate of adaptive introgression into species 2. The consequence of the feedback is, therefore, that the species that overcomes disease burden sooner exerts an additional disease pressure on the other species during the destabilization phase.

**Alternative model.** The model presented above focuses on behavioral responses to disease burden. However, given that the model requires a seemingly complex behavior—a population

must consciously or unconsciously modulate its interaction with other populations based on the amount of disease burden it experiences—it is worthwhile considering whether alternative models that emphasize other factors can produce similar dynamics.

We explored an alternative model, which focuses on demographic processes and endemic diseases. In this 'single-time-scale model', disease dynamics, adaptive introgression, and population dynamics occur in parallel, over a single time scale, as described by Eqs. (16)–(22). Here, we assume that the entire population of each species is infected by a suite of endemic pathogens, to which it has evolved tolerance. The pathogens inflict no harm on the hosts in the source species, but they are novel to the other species, and therefore cause increased mortality $\alpha_i(t)$ among infected individuals in the non-source species, who number $I_i(t)$. In this model, population densities, $N_i(t)$, are determined by density-dependent growth rates and by mortality rates (Eqs. (17) and (18)). We assume that between-species and within-species encounters are determined according to a process similar to Brownian motion of colliding particles, and therefore, contact intensities are modeled as proportional to the population densities (Eqs. (19) and (20)). In this model, response to disease burden is determined purely through demographic effects. Adaptive introgression has the effect of decreasing the mortality rates associated with the novel pathogens, and the introgression rates, like the disease transmission rates, are proportional to the inter-species contact intensity (Eqs. (21) and (22)).

Under symmetric initial conditions, the dynamics are similar to those observed in the two-time-scales model: an initial phase of high disease burden resulting in a decrease in population densities (Fig. 5, orange phase), followed by a long phase of low but stable disease burden with low population densities (Fig. 5, green phase), and eventual destabilization and population growth as disease burden is overcome (Fig. 5, blue phase).

When the initial conditions are asymmetric with respect to the pathogen packages, $\alpha_2(0) > \alpha_1(0)$, species 2 suffers higher mortality than species 1 due to novel pathogens. In this case, we observe that the two species initially respond strongly to disease burden and that population densities are reduced (Fig. 5d). A long stable phase of low population densities then follows. As is seen in the two-time-scales model, species 1 is released from disease burden sooner than species 2, when it begins to increase in density towards the initial density at the time of contact (Fig. 5e–g). With initial asymmetry in population densities ($N_2(0) > N_1(0)$), but equal pathogen packages ($\alpha_1(0) = \alpha_2(0)$), dynamics are qualitatively similar to those with asymmetry in pathogen packages, with the initially less dense species 1 overcoming disease burden sooner in the scenarios examined (Supplementary Note 1).

In the two-time-scales model, the separation of the ecological and evolutionary time scales simplifies the dynamics, since the dynamics in the ecological time scale are connected to the evolutionary time scale through the single summarizing parameter of disease impact, $F_i(t)$. In the single-time-scale model, however, interaction between disease, introgression, and demography occurs on the same time scale, resulting in more complex feedback among the different processes. Therefore, whether the consequence of the feedback is increased or decreased asymmetry at the time that species 1 overcomes disease burden depends on the source of the asymmetry and the parametrization of the model. In the scenario in Fig. 5, and with many of the other scenarios we explored (Supplementary Note 1), adaptive introgression in species 1 is more rapid than in species 2, and the differences between the disease-related mortality rates are increased at the time that species 1 overcomes disease burden,

relative to the initial asymmetry; however, unlike in the two-time-scales model, this outcome does not always arise, and in some cases, the differences in mortality rates between the species are lower at the time species 1 overcomes disease burden than at the beginning of the dynamics (e.g., Supplementary Figs. 8K and 10K). Once species 1 is released from disease burden and increases in size, the feedback is similar to that in the two-time-scales model, since the increased population densities cause higher disease transmission in the two-species system, which at this point is detrimental only to species 2.

**Interpretation of the models.** Our results demonstrate that disease and introgression dynamics can explain the persistent stable phase of inter-species dynamics that preceded the replacement of Neanderthals by Moderns. They also suggest that such dynamics may have implications regarding the replacement phase.

In the two-time-scales model, the adaptive response is in the contact rate itself, which is akin to assuming conscious or unconscious behavioral or cultural modification of contact rates[49,50]; for example, a band of hunter–gatherers may decrease inter-species contact by not accepting, or not taking by force, individuals from bands of the other species. They may limit contact even further by avoiding other bands completely. However, reduction in contact rates in response to disease burden may also have occurred due to decreased population densities, and therefore this model may be viewed as one that implicitly incorporates the combined effect of both behavior and demography.

The model includes an implicit assumption regarding intra-species gene flow. Immune-related alleles are assumed to spread rapidly once introgressed; that is, introgression that relieves disease burden acts on each species as if it were a cohesive unit. Although this assumption may be inappropriate for large, geographically structured, populations, our model is concerned only with the peripheral region in which interaction occurs, and in which selection could have acted to favor beneficial immune-related genes.

We also assume that genetic adaptation via de novo mutations is negligible compared to adaptive introgression in reducing disease burden. Under the scenario of Neanderthal–Modern interaction, this assumption amounts to assuming that introgression of pre-adapted alleles through interbreeding occurs more frequently than the appearance of novel mutations conveying resistance to a disease. We explore incorporation of regular adaptation into the models in Supplementary Note 2, with results qualitatively similar to those presented in Figs. 3–5, but with shorter stable phases.

In our alternative single-time-scale model, the adaptive response is a reduction in the level of disease-induced mortality, which is akin to assuming evolution of tolerance to the invading pathogen package. These exclusively demographic, rather than behavioral, dynamics can be readily interpreted in the context of an immunological or physiological response to assaults from novel pathogens. The model also assumes rapid within-species assimilation of acquired tolerance, similar to the assumption in the two-time-scales model. Modeling of regular adaptation in the single-time-scale model is addressed in Supplementary Note 2. Competition for resources between the species in the contact zone may have demographic consequences, but is not incorporated into the model; however, we have added competition to our model in Supplementary Note 3, and our analysis again yields similar results to those in Fig. 5, but with longer stable phases.

We have shown here that disease dynamics are sufficient to explain the long period in which the contact zone was confined to

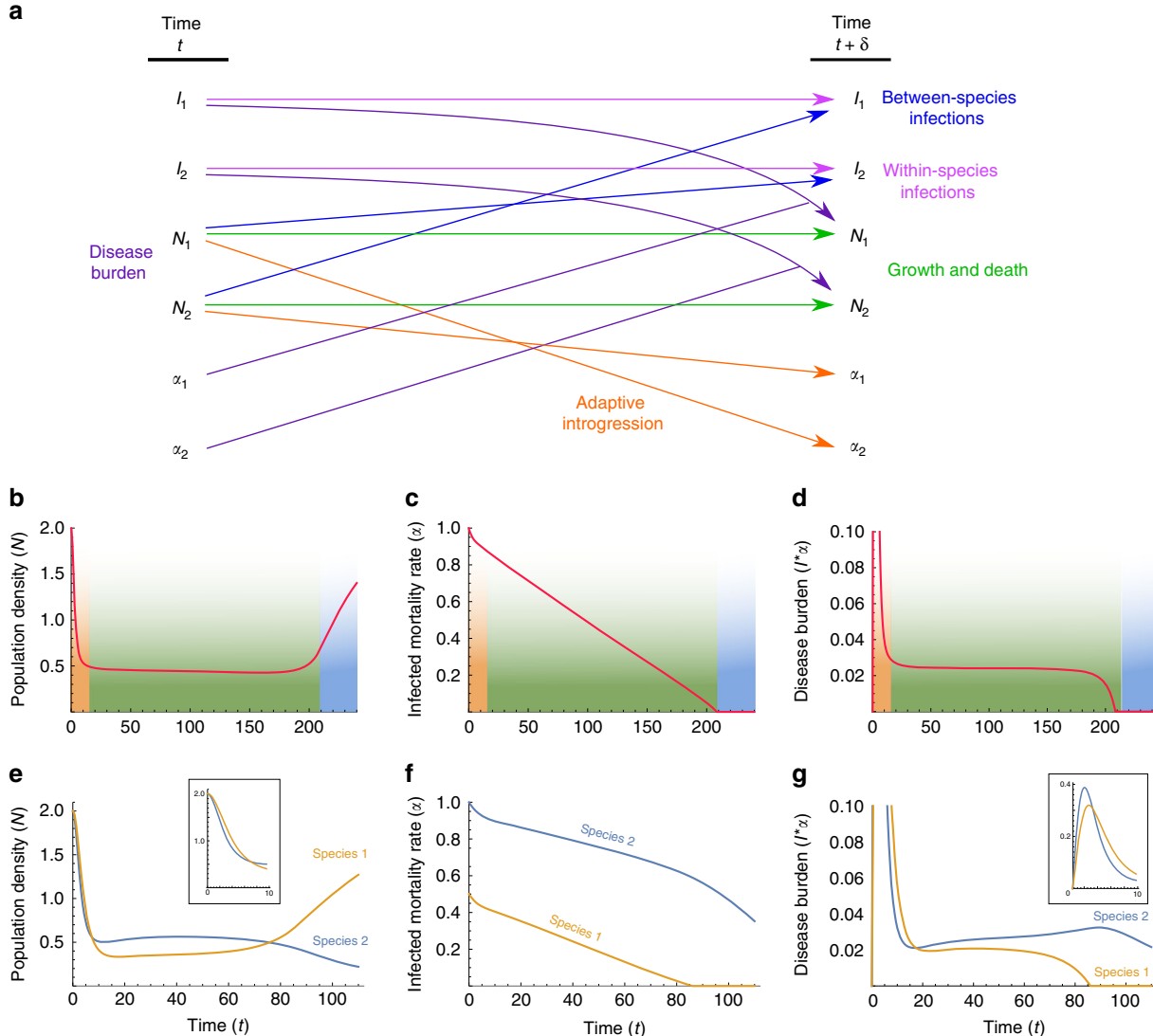

**Fig. 5** Alternative single-time-scale model of disease-introgression dynamics. **a** Schematic description of the model. The dynamics of the model are described by Eqs. (16)–(22). In this model, the population densities $N_i(t)$ are modeled, with disease burden caused by disease-induced mortality $\alpha_i(t)$ of infected individuals $I_i(t)$. Adaptive introgression reduces disease-induced mortality over time. (**b–d**) show the dynamics with symmetric initial conditions, comparable to Fig. 3, and (**e–g**) show results with asymmetric initial conditions, comparable to Fig. 4. (**b**) and (**e**) show population densities, $N_i(t)$ (Eqs. (17) and (18)). (**c**) and (**f**) show the added mortality rate due to infection from novel pathogens, $\alpha_i(t)$ (Eqs. (21) and (22)). (**d**) and (**g**) show disease burden, measured as $I_i(t) \times \alpha_i(t)$ (Eqs. (19)–(22)). The parameters for the scenario modeled are $\lambda = 0.15, K = 1, \beta_{11} = \beta_{22} = 0.5, \beta_a = 0.1, \mu = 0.05, c = 0.1$. For the symmetric case, (**b–d**), $\alpha_1(0) = \alpha_2(0) = 1$, and for the asymmetric case, **e–g**, species 1 experiences less disease-related mortality than species 2, $\alpha_1(0) = 0.5$ and $\alpha_2(0) = 1$. Inset graphs show the initial time periods in (**e**) and (**g**)

the Levant; however, we have not considered other factors that may have played a role. One such factor may be the species' adaptation to their respective environments at the core of their geographical ranges[7,34]. Many such adaptations have been proposed, ranging from morphology supporting faster or slower heat loss[37,51] to physiological traits that support different hunting methods[39]. Such adaptations could have limited migration of bands of one species into regions occupied by the other species, similarly to the effect of disease burden. However, local adaptation by itself would not explain the rapid destabilization of the interaction front. The period of 50–40 kya was not characterized by unprecedented environmental change that would have provided Moderns an advantage across the entirety of the Neanderthals' range[52–55] (but see refs. [56,57] and the discussion in ref. [58]). Similarly, no clear evidence has been found so far of a cultural shift among Moderns that would have provided them a sudden advantage over the Neanderthals[59].

**Implications for the replacement of Neanderthals by modern humans**. When the initial conditions are asymmetric, one species appears to acquire tolerance to novel diseases sooner than the other (Figs. 4 and 5d–f), permitting that species to expand its range earlier. A particularly plausible source of asymmetry in the case of modern humans and Neanderthals is the difference in pathogen complexes to which each of the species was adapted. Biotic diversity, on many taxonomic scales, is higher in the tropics[60], including in human pathogens[43,44]. In the Levant, where climate was intermediate between the temperate and tropical zones[61], many pathogens carried by both species would have been potential sources for diseases. It is therefore possible that Neanderthals would have had to adapt to a larger number of pathogens than did Moderns ($P_2(0) > P_1(0)$ or $\alpha_2(0) > \alpha_1(0)$ in our models, with species 1 representing Moderns and species 2 representing Neanderthals), leading to earlier Modern release from disease burden (Figs. 4 and 5e–g). Then, as modern humans

recovered from disease burden and expanded further into Eurasia, they might have encountered bands of Neanderthals whose lineages had not interacted with Moderns, and who had been far enough from the long-standing front of interaction not to have benefited from immune-related adaptive introgressions. These bands would thus have been even more vulnerable than Neanderthals in the Levant to pathogens carried by Moderns.

The scenario is analogous to more recent events, such as when Europeans arrived in the Americas in the 15th and 16th centuries with a more potent pathogen package than that of the local inhabitants, not because of climatic diferences, but because of higher population densities and contact with domesticated animals[48]. The colonization of the Americas was followed by rapid replacement of Native Americans, facilitated by disease spread[45–48].

Alternatively, a long-standing hypothesis argues that a genetic diversity difference upon contact could produce significant epidemics in the less-diverse population, due to immunological consequences of reduced diversity in that population's MHC region[46]. According to this hypothesis, asymmetry in susceptibility to pathogens may also have been a result of differences between the species in genetic diversities. Ancient genomes suggest that Moderns' genetic diversity exceeded that of Neanderthals'[62,63], in which case modern human populations might have been less susceptible to Neanderthal diseases than Neanderthal populations were to modern human diseases. This asymmetry could then produce the same asymmetries in model parameters described above for pathogen diversities.

Analogously, in animals, severe asymmetric epidemics can be caused by contact between closely related species that have been separated for long periods[64]. For example, an ongoing epidemic of squirrelpox in Eurasian red squirrels in the UK and Ireland was most likely introduced by gray squirrels, which are native to North America but invasive in Europe[65,66]. Gray squirrels are largely tolerant to squirrelpox, having co-evolved with the virus in North America, but the disease is almost always fatal to red squirrels[67]. Consequently, red squirrel populations are currently in significant decline, and they are often replaced by gray squirrels[65]. In other examples of contact between closely related animal species, disease transmission may produce some of the elements of our scenario: reinforcement of a narrow contact zone, erosion of the contact zone, and promotion of hybridizations between the species[64].

That diseases played an important role in the inter-species dynamics of modern humans and Neanderthals[31,32,41,42] is suggested by studies that compare Neanderthal genomes with current-day Modern genomes, and that argue that genomic regions relating to disease immunity and tolerance are enriched in introgression of Neanderthal genes[2,22,26,28,29,31]. This result suggests that introgression was adaptive, and that diseases were a significant enough burden that natural selection in Moderns favored introgressed lineages that included immune-related genes. Interestingly, recent indications from analysis of European Neanderthal genomes suggest that gene flow was not symmetric between the species, and that more genes were introgressed from Neanderthals into Moderns than in the other direction[63]. This asymmetry is predicted by our two-time-scale model, which includes differential inter-species contact intensities (Fig. 4a, dashed curves); note, however, that the Neanderthals analyzed in producing this evidence of asymmetry were not sampled in the Levant[63].

**Future investigation of the Neanderthal–Modern disease landscape**. Further investigation of the role of diseases in the interaction between Neanderthals and modern humans would

require better understanding of the pathogen landscape during this period. One direction is the study of the genomic regions that were under selection at the time of contact[2,22,26,28,29,31]. A few recent phylogenetic studies of pathogens have indicated that some pathogens might have been transmitted from Neanderthals or other archaic humans to Moderns[24–27]. A particularly useful direction for elucidating the disease landscape would be genomic studies[23,68–72] of ancient pathogens recovered from archeological Neanderthal and modern human sites, since they would potentially provide more direct opportunities to evaluate the disease burden that the two species experienced, and occurrences of inter-species disease transmissions, at different times.

The consequences of inter-species disease dynamics may be evident in archeological findings as well. For example, as suggested by our model, disease burden in the Levant might have affected population densities (Figs. 3a, 4a, 5b and 5g), reducing them compared to those in adjacent regions where novel pathogens had not yet been introduced, to those prior to contact, or to those after release from the disease burden. Population densities might be estimated by the assessment of archeological site density and site complexity[73,74]. Population density can potentially also be assessed via analysis of resource exploitation; for example, mean prey size may reflect predation pressure, allowing estimation of hominin population density[75,76]. Further archeological studies that target parameters of population demography in the Levant at the time in question will be important for testing the predictions of our models.

We note that our models assume a simple overall demographic history, with a single contact zone followed by further expansion of modern humans into Eurasia. It is possible to extend the models to consider more complex scenarios, such as multiple and intermittent contact zones[36] or several expansions across Eurasia[77]. As more data concerning possible demographic scenarios become available, it will be increasingly possible to assess the extent to which the conclusions from our models hold for these more complex scenarios.

We have not modeled spatial structure in the two species. Incorporating spatial structure would require assumptions about: (1) the spatial extent of a contact zone in which the two species could be regarded as well-mixed; (2) the relative densities of the two species in this well-mixed zone; (3) the rate at which the relative densities of the species decreased as a function of distance from the contact zone; and (4) the rates at which genetic introgressions in the contact zone spread to other regions. Incorporating this spatial structure could assist in examining how vulnerability to pathogens was distributed across Eurasia, and how it could have shaped the interactions between the species during the replacement of Neanderthals by modern humans.

**Conclusion**. A major focus in the study of the inter-species dynamics between modern humans and Neanderthals has been the relative rapidity with which Moderns replaced Neanderthals across the majority of Eurasia. In this study, we suggest that analyzing the phase that preceded the eventual replacement is valuable as well. That the two species' front of interaction was constrained to the Levant for tens of thousands of years is puzzling, particularly in light of the short time—a few millennia—within which the replacement across the rest of Eurasia was completed. We have drawn insights from the field of disease ecology to suggest that infectious disease dynamics may explain the long period of stability that preceded the replacement. We have explored this possibility using mathematical modeling, deriving predictions that can inform future exploration. We propose that this approach provides insights into the inter-species dynamics at the transition between the Middle and Upper

Paleolithic periods, particularly due to the sparsity of the material record from this period and in consideration of the promise that DNA sequencing and dating technologies hold. Such modeling, coupled with new technologies and with novel approaches in prehistoric archeology, may act synergistically to allow a new interpretation of this exciting period in human evolution.

## Methods

**Modeling approach**. We develop a modeling approach to explore the effects of disease and introgression in a two-species system. In the two-time-scales model, the evolutionary and ecological times scales are separated (see refs. [78–80] for other such models). On the ecological time scale, the spread of diseases is modeled, and on the evolutionary time scale, the species' response to disease burden and the effects of introgression are modeled (Fig. 2). We assume that each species initially carries a pathogen package with which it had co-evolved, and to which it is therefore tolerant. A species suffers no significant negative impact from its associated pathogen package, but its pathogens are novel to the other species, which is vulnerable to them and is harmed considerably.

In the single-time-scale model, a similar scenario is modeled, and a single time scale is used to model ecology, disease, and introgression as interacting parallel processes. The process that drives the response to disease burden is a demographic process.

**Two-time-scales model**. On the evolutionary time scale, we model disease burden, response to disease burden, and introgression using a discrete-time model (Fig. 2). Each time step in this model ($t$) is assumed to be large enough for disease processes to be experienced by the populations and for the populations to respond to these disease pressures. For simplicity, we focus on disease dynamics described by within-species and between-species contact rates ($\beta_{ij}$ for contact rates in which pathogens from species $i$ are transmitted to species $j$). We model response to disease burden implicitly as adjustments of these rates.

Initially, each species $i$ experiences a novel pathogen package of size $P_i(0)$. This package size can be interpreted as the number, or diversity, of novel pathogens in species $j$ ($j \neq i$) to which species $i$ is still vulnerable, the genes providing immunity to these pathogens not having yet introgressed into species $i$. At each time $t$, the disease burden experienced by the pair of species is modeled as

$$D_1(t) = P_1(t)F_1\big(\beta_{11}(t), \beta_{12}(t), \beta_{21}(t), \beta_{22}(t)\big) \qquad (1)$$

$$D_2(t) = P_2(t)F_2\big(\beta_{11}(t), \beta_{12}(t), \beta_{21}(t), \beta_{22}(t)\big), \qquad (2)$$

where $D_i(t)$ is the disease burden experienced by species $i$ at time $t$, and $F_i$ is a disease-ecology model describing the average impact of a single pathogen on population $i$ given the contact rates $\beta_{ij}(t)$. $F_i$ describes the faster ecological time scale in the model.

We model the ecological process using a two-species well-mixed SIR epidemic model[81,82] (Fig. 2b). In an SIR model, individuals can be in one of three states—susceptible ($S$), infectious ($I$), and recovered/removed ($R$), which are measured in terms of their proportion in the entire population (i.e. $S + I + R = 1$). In addition to transmission rates ($\beta_{ij}$), the SIR model requires additional parameters describing the recovery/removal rates for the two species, $\gamma_1$ and $\gamma_2$ for species 1 and 2, respectively; these parameters represent rates of either recovery with immunity or death from the disease, which have similar outcomes in their effect on spread of the disease, because in both cases the host can no longer transmit the disease. For simplicity, we assume symmetry between the species in recovery/removal rates, i.e. $\gamma = \gamma_1 = \gamma_2$. The equations governing the dynamics of the SIR model over time, which we term $\tau$ to distinguish from the longer evolutionary time scale $t$, are:

$$\frac{dS_1(\tau)}{d\tau} = -\beta_{11}S_1(\tau)I_1(\tau) - \beta_{21}S_1(\tau)I_2(\tau) \qquad (3)$$

$$\frac{dS_2(\tau)}{d\tau} = -\beta_{22}S_2(\tau)I_2(\tau) - \beta_{12}S_2(\tau)I_1(\tau) \qquad (4)$$

$$\frac{dI_1(\tau)}{d\tau} = \beta_{11}S_1(\tau)I_1(\tau) + \beta_{21}S_1(\tau)I_2(\tau) - \gamma I_1(\tau) \qquad (5)$$

$$\frac{dI_2(\tau)}{d\tau} = \beta_{22}S_2(\tau)I_2(\tau) + \beta_{12}S_2(\tau)I_1(\tau) - \gamma I_2(\tau) \qquad (6)$$

$$\frac{dR_1(\tau)}{d\tau} = \gamma I_1(\tau) \qquad (7)$$

$$\frac{dR_2(\tau)}{d\tau} = \gamma I_2(\tau). \qquad (8)$$

Note that at this time scale the $\beta_{ij}$ parameters are considered fixed, and they change only at the evolutionary time scale. Individuals of species $i$ in the susceptible state, $S_i$, can transition to the infected state $I_i$ by being infected either by individuals from their own species, at rate proportional both to $\beta_{ii}$ and to the proportion of contacts

between susceptible and infected individuals, or, similarly, by being infected by individuals from the other species at a rate proportional to both $\beta_{ji}$ and the proportion of inter-species contacts between infected and uninfected individuals. Infected individuals transition to the recovered/removed state at a fixed rate, $\gamma$.

Under this SIR model with specific values of $\beta_{ij}$ and $\gamma$, the impact of an average epidemic wave on a species, $F_i$ for species $i$, is measured by the proportion of the population that was infected during an entire run of the epidemic:

$$F_i = \lim_{\tau \to \infty} R_i(\tau). \qquad (9)$$

We measure this proportion in the case that the epidemic originates in relatively few individuals in species $j$ ($I_i(0) = 0$ and $I_j(0) = 0.01$). The limit is taken of $R$ since infected individuals inevitably end up in the $R$ state, and therefore, tracking the eventual number of recovered/removed individuals amounts to tracking the number of overall infected individuals. The scenario modeled using an SIR model at the ecological level is one in which a species experiences an epidemic wave originating in the other species. The epidemic spreads in the combined system of the two species, but only the non-source species is vulnerable to the effects of the disease. Consequently, the spread of disease, in our model, has no impact on the source species, except in its effect on disease spread in the non-source species.

On the evolutionary time scale, after experiencing disease burden (e.g. repeated epidemic outbreaks that originated in the other species), the species modify their contact rates accordingly. Such modifications can be behavioral in nature, either via a conscious process, whereby individuals actively notice that reducing within-species and particularly between-species contact rates reduces the impact of outbreaks, or via unconscious alteration of behavior, involving long-term selection on cultural traits, or instinctive factors, such as stress-induced aversion to strangers. Contact rate modification can also be demographic in nature, where disease burden reduces population densities and increases the geographic distances between groups of individuals.

Each species is also assumed to have initial intrinsic within-species and between-species contact rates, which reflect the typical contact rates unhindered by the novel pathogens. We model the adjustment of contact rates as being proportional to the disease burden, countered by the tendency of species to return back to their intrinsic behavioral and demographic states. These tendencies are modeled as proportional to the difference between the current state and the original state. Therefore, adjustment of contact rates in response to disease burden is modeled as follows:

$$\beta_{11}(t) = \beta_{11}(t-1) - a_{11}D_1(t-1) + b_{11}\big[\beta_{11}(0) - \beta_{11}(t-1)\big] \qquad (10)$$

$$\beta_{22}(t) = \beta_{22}(t-1) - a_{22}D_2(t-1) + b_{22}\big[\beta_{22}(0) - \beta_{22}(t-1)\big] \qquad (11)$$

$$\beta_{12}(t) = \beta_{12}(t-1) - a_{12}D_2(t-1) + b_{12}\big[\beta_{12}(0) - \beta_{12}(t-1)\big] \qquad (12)$$

$$\beta_{21}(t) = \beta_{21}(t-1) - a_{21}D_1(t-1) + b_{21}\big[\beta_{21}(0) - \beta_{21}(t-1)\big], \qquad (13)$$

where the $a_{ij}$ are parameters scaling the response to disease burden and the $b_{ij}$ are parameters scaling the tendency to return to the initial contact rates. In our model, because disease burden is eventually removed ($D_i(t) \to 0$), the contact rates tend to return to their initial state ($\beta_{ij}(t) \to \beta_{ij}(0)$). For simplicity, we assume symmetry for these parameters, and define $a = a_{ij}$ and $b = b_{ij}$ for all $i$ and $j$.

Finally, we model the effect of introgression on the number of novel pathogens experienced by the species, assuming that it is proportional to inter-species disease transmission:

$$P_1(t) = \max\big\{P_1(t-1) - c\,\beta_{21}(t-1), 0\big\} \qquad (14)$$

$$P_2(t) = \max\big\{P_2(t-1) - c\,\beta_{12}(t-1), 0\big\}. \qquad (15)$$

Here, $c$ is a parameter scaling the rate of adaptive introgression relative to the rate of disease transmission. Since these two rates are defined at different time scales (disease transmission on the ecological time scale in Eqs. (10)–(13), and introgression on the evolutionary time scale in Eqs. (14) and (15)), $c$ has an additional role in the models as the parameter that scales the two time scales.

In Eqs. (10)–(13), the response due to disease burden is modeled as a reduction in within-species and incoming between-species rates. This choice is particularly appropriate for behavioral responses to disease pressure, where between-species contacts are most likely determined by the willingness of the receiving population to accept contact.

**Single-time-scale model**. In this section, we describe a continuous-time model of an endemic disease process that occurs in parallel to immune-related introgression and demographic dynamics. The model is a birth–death SI model[82] in which population sizes are affected by the birth and death rates, and disease burden is modeled as an increased death rate $\alpha_i(t)$ in individuals infected by novel pathogens over the natural death rate ($\mu$). The model tracks the species population densities ($N_i(t)$) given intrinsic growth rates ($\lambda$). In order for growth rates to be density-

dependent, the actual growth rates are given as a function of $N_i$[83,84]:

$$\Lambda(N_i) = \frac{\lambda K}{K + N_i}. \tag{16}$$

Here, $\lambda$ is the maximal growth rate (achieved in sparse populations), and $K$ is a half-maximum growth rate density parameter. In other words, $\Lambda(K) = \frac{\lambda}{2}$; populations of density $K$ grow at half the maximum growth rate. For simplicity, we assume that the intrinsic growth rates ($\lambda$), the half-maximum growth rate density parameters ($K$), and the natural mortality rates ($\mu$), are the same for both species.

Population dynamics therefore reflect population growth, natural population mortality, and the additional mortality incurred due to the novel pathogens, where the number of individuals infected by novel pathogens is $I_i(t)$:

$$\frac{dN_1(t)}{dt} = \frac{\lambda K}{K + N_1(t)} N_1(t) - \mu N_1(t) - \alpha_1(t) I_1(t) \tag{17}$$

$$\frac{dN_2(t)}{dt} = \frac{\lambda K}{K + N_2(t)} N_2(t) - \mu N_2(t) - \alpha_2(t) I_2(t). \tag{18}$$

Initially, prior to inter-species contact, we assume no individuals are infected by novel pathogens ($I_i(0) = 0$), and that the populations are at demographic equilibrium, implying that $N_i(0) = K \frac{\lambda - \mu}{\mu}$ (see Supplementary Note 1 for different assumptions).

Next, we model disease dynamics in the populations using an SI model, in which each individual can only move from a susceptible state to an infected state, and cannot recover. The susceptible individuals ($N_i(t) - I_i(t)$ in this model) can be infected either by any individual from the other species, or by infected individuals in its own species. Note that we assume that all individuals in the source population are infected; this assumption is plausible for an endemic pathogen that has co-evolved with the species, and for which that species has little physiological response. Under the SI model, infected individuals can be removed only when they die, at rate $\mu + \alpha_i(t)$. The dynamics of the disease are therefore modeled by

$$\frac{dI_1(t)}{dt} = [N_1(t) - I_1(t)][\beta_a N_2(t) + \beta_{11} I_1(t)] - [\mu + \alpha_1(t)] I_1(t) \tag{19}$$

$$\frac{dI_2(t)}{dt} = [N_2(t) - I_2(t)][\beta_a N_1(t) + \beta_{22} I_2(t)] - [\mu + \alpha_2(t)] I_2(t), \tag{20}$$

where $\beta_{11}$ and $\beta_{22}$ are the within-species transmission parameters and $\beta_a$ is the symmetric between-species transmission parameter. These transmission parameters are a composite product of, and are proportional to, the contact rates (the rate at which individuals are close enough together to facilitate a pathogen transmission event) and the probability of transmission per contact[85]; for simplicity, we refer to these parameters as contact rates, since we assume that the probability of transmission per contact is fixed. Note that here, inter-species transmission is modeled as being driven purely by random encounters, affected only by the population densities. Hence, there is a single parameter for inter-species transmission ($\beta_a$), unlike in the two-time-scales model, which has one for each direction of transmission ($\beta_{ij} \neq \beta_{ji}$ for $i \neq j$). Additionally, contact rates in this model are fixed (i.e. they do not change throughout the dynamics) and are not subject to behavioral alteration in response to disease burden.

Finally, we describe the effect of transmission between the species on introgression. Here, disease burden is modeled not as a function of the number of pathogens, but more implicitly as the cumulative effect expressed by increased mortality rate ($\alpha_i$). Therefore, we model immune-related introgression as reducing the mortality associated with being infected by the novel pathogens, factored by the transmission intensity (force of transmission) between the species:

$$\frac{d\alpha_1(t)}{dt} = -c\beta_a N_2(t) \text{ provided } \alpha_1(t) > 0, \text{ otherwise } \frac{d\alpha_1(t)}{dt} = 0 \tag{21}$$

$$\frac{d\alpha_2(t)}{dt} = -c\beta_a N_1(t) \text{ provided } \alpha_2(t) > 0, \text{ otherwise } \frac{d\alpha_2(t)}{dt} = 0. \tag{22}$$

$c$ is a scaling parameter interpreted as the ratio between immune-related gene transmission and disease transmission.

Equations (17)–(22) describe a simple SI model of endemic diseases in two species, where response to disease burden is purely demographic, and in which contact rates between the species are not affected by disease burden.

**Numerical analysis**. All systems of partial differential equations were solved numerically using the Mathematica software[86]. These numerical solutions were obtained deterministically from explicitly stated initial conditions.

**Reporting summary**. Further information on research design is available in the Nature Research Reporting Summary linked to this article.

## Data availability
No datasets were generated or analyzed in this study.

## Code availability
The code for generating the numerical solutions described in this paper can be downloaded as a Mathematica notebook from github.com/GiliG/Neanderthal_disease_and_introgression.

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

## Acknowledgements

We would like to thank David Gokhman, David Freisem, David Enard, and Anna Belfer-Cohen for insightful comments. G.G. and O.K. were funded by fellowships from the Stanford Center for Computational, Evolutionary, and Human Genomics (CEHG). O.K. was also funded by a grant from the John Templeton Foundation. This project was supported by NSF grant BCS-1515127 awarded to NAR.

## Author contributions

G.G. and O.K. conceived the study. G.G., W.M.G. and O.K. developed the models. N.A.R., M.W.F. and E.H. contributed ideas to the conceptual framing of the project. G.G., O.K. and W.M.G. drafted the paper. All authors read and commented on the manuscript.

## Competing interests

The authors declare no competing interests.
