## [Peer Review File · Nature Communications]

Reviewers' Comments:

Reviewer #1:

Remarks to the Author:

This is a compelling paper that develops and tests models to explore the role of AMH-Neanderthal interaction in relation to disease burden, asymmetric responses, and hominin species replacement in the Levant (which differs considerably from that of the rest of Eurasia). The authors highlight the potential for asymmetric responses to pathogen packages and this in turn raises important questions for how we think about hominin contact and replacement beyond the initial or first point of contact between groups.

The main result that disease and introgression dynamics can explain the persistent stable phase of inter-species dynamics that preceded the replacement of Neanderthals by AMH is compelling and supports the differential pathogen resistance model. The results of this paper are extremely important and will help to deepen our understanding of the role of disease as a human selective pressure. One of the main strengths of the approach of the models used in the paper is that it places the focus on human species as part of the wider eco-evo adaptive environment rather than as culturally endowed beings who exist alongside but beyond somehow above it.

The authors make the important point that 'further investigation of the role of diseases in the interaction between Neanderthals and Moderns would require a better understanding of the pathogen landscape during this period'. This paper, in combination with others in the same vein, will set important diagnostic criteria for better understanding the data from ancient human genomes and phylogenetic studies of pathogens.

The methods are extremely clearly laid out, and allow for repetition (and crucially for expansion with new data), and as the authors suggest has potential for developing predictions for identification of new sites etc.

The call to not model spatial structures is sound and is a layer of complexity that can be revisited as more arch data comes to light. The point about the severity of exposure to pathogens increasing as distance increases from the zone of adaptive-introgression is well made but perhaps it might be worth making this slightly punchier in relation to the decision to exclude spatial structures from the models. Is the analogy to European contact with the Americas the best one? Perhaps animal examples might be more informative? Squirrel pox carried by Grey Squirrels and its impact on Red Squirrels is one such example that might be worth considering...

Ultimately this is an extremely thought-provoking and impressively executed paper that will be of great interest to those working across the field.

Dr Simon Underdown
Oxford Brookes University, UK

Reviewer #2:

Remarks to the Author:

This paper is a very interesting contribution, but as I am not a specialist in the modelling methods, I will primarily comment on the interpretation and implications of the models.

Greenbaum and colleagues address a question that has been in the minds of most

palaeoanthropologists for decades - how did the contacts between Neanderthals and the first modern humans coming into their range look, especially taking into account a likely extended coexistence in some areas.

Using mathematical models from ecology the authors explore how disease transmission could have influenced the contact zone between the two groups, slowing down range expansion. Adaptive introgression allowed the groups to develop resistance to each other's pathogens, and spread further.

The mechanisms the authors propose fit well with the genetic evidence, especially the large amounts of adaptive introgression (both from Neanderthals and Denisovans) in areas of the genome linked to immune response. Clearly, disease resistance played a major role in how the contacts between different hominin groups played out.

There are a few things that should be addressed before publication:

One important difference between Neanderthals and modern humans (I don't really care for the "Moderns" terminology used by the authors) is population connectivity and genetic diversity. Most of the genetic evidence from Neanderthals indicates that they lived in small, relatively inbred groups - best evidence for this is from the Altai Neanderthal, but small N_e is also seen in El Sidron, Vindija and Mezmaiskaya. In contrast, the earliest modern humans we have DNA from (Ust' Ishim, Sungir) seem to be more diverse, and in the case of Sungir Sikora et al. (2016) proposed that there were social mechanisms of exogamy.

It would be useful if the authors could discuss how the low genetic diversity might have influenced the dynamics between the two groups.

I also think that their Figure 1 is a bit too simplified, there is more and more evidence about the importance of the Arabian peninsula from modern human dispersal, and so the contact zone between Neanderthals and modern humans might have been significantly larger than described here. This does not alter the conclusions of the paper though.

There is also quite a bit of discussion around the question whether there was actually a long standstill and coexistence of Neanderthals and modern humans in the Levante - people have also argued that there has been more of a back and forth, perhaps related to climatic and environmental changes (e.g. several papers from the 1998 Akazawa volume). It would be useful to discuss this a bit more in depth.

There are also a few typos in the text that would benefit from a bit more cleaning up.

All in all though this is a thoughtful and interesting contribution that is definitely worth publishing once these minor concerns have been addressed.

Bence Viola, Dept. of Anthropology, U. of Toronto

Reviewer #3:

Remarks to the Author:

This is a very interesting paper with, I think, plausible models for the interactions of modern humans and Neanderthals across contact zones, and the potential roles of pathogens on those interactions. I have marked up some small points on the main manuscript (although I left American spellings for words like behaviour and Palaeolithic), and I have no competence to comment on the modelling methods and results. My comments centre on the general approach and the context of the questions

that are addressed. I also note below some other literature that might be relevant here.

My main point is that this paper assumes that there was a long-lasting and localised inter-species boundary between moderns and Neanderthals in the Levant. There may well have been, but it is by no means certain that this boundary occurred uniquely in the Levant rather than, say, Arabia or Iraq, nor is there yet adequate chronological control to even show that these populations actually overlapped in the Levant at any one time, although it is likely they did, of course. These limitations/qualifications need to be admitted near the beginning of the paper, before proceeding to the assumptions that are used for the modelling.

Another assumption seems to be that moderns did not extend beyond the Levant until destabilisation of the interaction front around 45-50 ka. However, there is already a modern human fossil at Ust-Ishim in Siberia directly dated at about 45 ka, with an estimated introgression date for its Neanderthal DNA of ~55 ka. And as discussed in, for example, Bae et al. *Science* 358, eaai9067 (2017), there is growing evidence that moderns extended beyond the Levant well before 50 ka, and probably before 80 ka, so how does that affect the arguments of this paper about disease constraints on early moderns in Eurasia? Indeed, there is a paper in press that seemingly takes *H. sapiens* beyond the Levant at an even earlier date. Additionally, genetic arguments have been advanced for an earlier interaction of the sapiens and Neanderthal lineages, as reflected by a probable Middle Pleistocene spread of mtDNA (Posth et al. *Nature Communications* volume 8, Article number: 16046 2017). Even if the authors question some of these findings or somehow consider them irrelevant to their analyses, I think they do need to at least discuss the data that suggest more complex scenarios than those that they model.

Finally, I was interested enough by this paper to investigate whether there is already a relevant literature for non-human species, and indeed there is. For example (in this case concentrating on parasites) Theodosopoulos et al. *Parasites and Host Species Barriers in Animal Hybrid Zones* 2018DOI:<https://doi.org/10.1016/j.tree.2018.09.011>

Species barriers are tested in hybrid zones when gene flow occurs between hybridizing species. Hybridization can erode species barriers, lead to the introgression of adaptive traits, or remain stable through time. Outcomes in hybrid zones are influenced by divergence between the hybridizing taxa, behavior, ecology, and geography. Parasites and pathogens play a major role in host fitness and appear to have varied impacts on species barriers in hybrid zones. We comprehensively reviewed the literature on parasitism in animal hybrid zones and present an evolutionary framework within which to consider parasite–hybrid interactions. Parasites most frequently show potential to contribute to species barrier breakdown in hybrid zones, but also frequently show potential to facilitate the maintenance of species barriers. Incorporating eco-immunology, parasite community theory, and spatiotemporal approaches will be important as genomic tools allow researchers to examine parasites and hybrid zones at greater resolution and in a diversity of natural habitats.

I do not require anonymity for this review - Chris Stringer

Reviewer #4:

Remarks to the Author:

Greenbaum et al present a modelling study of interactions between modern humans and Neanderthals in the Levant, in the period before the out of Africa episode of modern humans that occurred 45 to 50 thousand years ago. Their study aims to explain the long period of stable coexistence of the two species in the Levant before the sudden modern humans spread toward Eurasia. The basic hypothesis is that each of the two species would have carried a package of pathogens from the environment in which they evolved (temperate for Neanderthals and tropical for modern humans), to which the other species would have been vulnerable. The models developed show that this hypothetical configuration well explains a long period of stable coexistence between the two species and that introgression of

resistant genes due to admixture between them would explain the gradual acquisition of resistance in each of the two species. Models show that if demographic and disease conditions are symmetrical, then neither species takes an advantage over another. However, if conditions are asymmetric, then one species can get rid of its disease burden faster than the other, and thus expand over the other species range. According to the results presented, an asymmetry in the pathogen packages could theoretically explain both the long period of contact between modern humans and Neanderthals in the Levant and the sudden expansion of modern humans to the detriment of Neanderthals.

I find the paper very well written, clear and the proposed hypothesis interesting and well supported by the literature cited. The proposed models seem robust, but I was not able to verify all its mathematical background.

I am thus in favor of publishing this article once the points below have been addressed.

My main concerns regard an aspect that I do not think sufficiently addressed in the paper: the densities of interacting populations.

First, the authors hypothesize that epidemics in Neanderthals and modern humans could explain the long period of contact between them in the Levant. However, were the densities of populations of these two species and the connectivity between them in the area sufficient to allow epidemics to spread? The densities of these two species are usually thought to be very low at this time (eg Bocquet-Appel et al, *Antiquity* 2000, *J Archaeol Sci* 2000). Would such low densities be in accordance with the model's assumptions?

Secondly, the results of the study show that the species with the lowest density will get rid of its disease burden the fastest and thus be able to spread further. However, it is generally assumed that modern human densities was higher than density of Neanderthals (eg Currat et al, *PNAS* 2011 and references above), which would contradict the results presented. We can also think that modern humans living in Africa constituted a reservoir for modern humans in the Levant, so their population would be larger than the Neanderthal population. Do the authors have arguments in favor of a lower density of modern humans than Neanderthals in the Levant?

In addition, in the second paragraph of page 9, there is a contradiction between the results of the first and second parts. In the first part, species 2 suffers from higher mortality (and therefore presumably lower density), but species 1 gets rid of its disease burden the fastest. This last result is in contrast to part 2 below where the less dense species 1 is the first to remove its burden.

Third, according to the model proposed by the authors, the expansion of humans in Eurasia would have occurred from the Levant, and not from Africa as supported by many studies (eg Prugnolle et al, *Current Biology* 2005). How can the authors explain this discrepancy?

I also noted a few edits:

- Page 2, end of the fourth paragraph: "suggest suggests", one "suggest" should be removed.
- Page 8, fifth paragraph: "single-time-scale model model", one "model" should be removed.
- Page 9, last paragraph: "taking by force, individuals of the other species", it is probably meant "population" instead of "species" in that context.

We thank the reviewers for their very thoughtful comments and suggestions. We have revised the manuscript accordingly, and we believe that it is much improved as a result.

Please find below a detailed response to each of the comments.

Reviewer #1

This is a compelling paper that develops and tests models to explore the role of AMH-Neanderthal interaction in relation to disease burden, asymmetric responses, and hominin species replacement in the Levant (which differs considerably from that of the rest of Eurasia). The authors highlight the potential for asymmetric responses to pathogen packages and this in turn raises important questions for how we think about hominin contact and replacement beyond the initial or first point of contact between groups.

The main result that disease and introgression dynamics can explain the persistent stable phase of inter-species dynamics that preceded the replacement of Neanderthals by AMH is compelling and supports the differential pathogen resistance model. The results of this paper are extremely important and will help to deepen our understanding of the role of disease as a human selective pressure. One of the main strengths of the approach of the models used in the paper is that it places the focus on human species as part of the wider eco-evo adaptive environment rather than as culturally endowed beings who exist alongside but beyond somehow above it.

The authors make the important point that ‘further investigation of the role of diseases in the interaction between Neanderthals and Moderns would require a better understanding of the pathogen landscape during this period’. This paper, in combination with others in the same vein, will set important diagnostic criteria for better understanding the data from ancient human genomes and phylogenetic studies of pathogens.

The methods are extremely clearly laid out, and allow for repetition (and crucially for expansion with new data), and as the authors suggest has potential for developing predictions for identification of new sites etc.

1. COMMENT: The call to not model spatial structures is sound and is a layer of complexity that can be revisited as more arch data comes to light. The point about the severity of exposure to pathogens increasing as distance increases from the zone of adaptive-introgression is well made but perhaps it might be worth making this slightly punchier in relation to the decision to exclude spatial structures from the models.

REPLY: Accepted. We have added a more thorough description of our modeling decisions regarding spatial structure, and how they relate to the increase in pathogen severity as a function of distances from the initial region of interaction, as suggested (lines 404-411).

2. COMMENT: Is the analogy to European contact with the America's the best one? Perhaps animal examples might be more informative? Squirrel pox carried by Grey Squirrels and its impact on Red Squirrels is one such example that might be worth considering...

REPLY: Thank you for this example. There are indeed interesting analogous non-human examples for which the dynamics we describe could be relevant, and from which we can possibly learn about dynamics that have occurred in the past. We have added this nice example in lines 355-361.

Ultimately this is an extremely thought-provoking and impressively executed paper that will be of great interest to those working across the field.

Reviewer #2

This paper is a very interesting contribution, but as I am not a specialist in the modelling methods, I will primarily comment on the interpretation and implications of the models.

Greenbaum and colleagues address a question that has been in the minds of most palaeoanthropologists for decades - how did the contacts between Neanderthals and the first modern humans coming into their range look, especially taking into account a likely extended coexistence in some areas.

Using mathematical models from ecology the authors explore how disease transmission could have influenced the contact zone between the two groups, slowing down range expansion. Adaptive introgression allowed the groups to develop resistance to each other's pathogens, and spread further.

The mechanisms the authors propose fit well with the genetic evidence, especially the large amounts of adaptive introgression (both from Neanderthals and Denisovans) in areas of the genome linked to immune response. Clearly, disease resistance played a major role in how the contacts between different hominin groups played out.

There are a few things that should be addressed before publication:

3. COMMENT: One important difference between Neanderthals and modern humans (I don't really care for the "Moderns" terminology used by the authors) is population connectivity and genetic diversity. Most of the genetic evidence from Neanderthals indicates that they lived in small, relatively inbred groups - best evidence for this is from the Altai Neanderthal, but small Ne is also seen in El Sidron, Vindija and Mezmaiskaya. In contrast, the earliest modern humans we have DNA from (Ust' Ishim, Sungir) seem to be more diverse, and in the case of Sungir Sikora et al. (2016) proposed that there were social mechanisms of exogamy. It would be useful if the authors could discuss how the low genetic diversity might have influenced the dynamics between the two groups.

REPLY: This is a very interesting idea, and we thank the reviewer for this nice suggestion. MHC genetic diversity may play an important role in susceptibility of populations to diseases. If the MHC diversity of modern humans was higher than that of Neanderthals, as suggested by the evidence mentioned by the reviewer, then modern humans could have been immune to more of the Neanderthal's diseases, effectively lowering the pathogen package they were exposed to. Therefore, higher genetic diversity in modern humans may have contributed to the asymmetry in disease packages we suggest in the paper. We have added this discussion in lines 346-354. Additionally, in response to the reviewer's comment, we try to use the abbreviated "Moderns" more sparingly.

4. COMMENT: I also think that their Figure 1 is a bit too simplified, there is more and more evidence about the importance of the Arabian peninsula from modern human dispersal, and so the contact zone between Neanderthals and modern humans might have been significantly larger than described here. This does not alter the conclusions of the paper though.

REPLY: Both the comment and the acknowledgement about the conclusions are well appreciated and accepted. We have added a comment about the Arabian Peninsula in the caption of Figure 1, and also in the main text in line 93.

5. COMMENT: There is also quite a bit of discussion around the question whether there was actually a long standstill and coexistence of Neanderthals and modern humans in the Levante - people have also argued that there has been more of a back and forth, perhaps related to climatic and environmental changes (e.g. several papers from the 1998 Akazawa volume). It would be useful to discuss this a bit more in depth.

REPLY: We have added a comment on this point in lines 91-93, and in lines 398-403 we discuss the impact of such dynamics on the models.

6. COMMENT: There are also a few typos in the text that would benefit from a bit more cleaning up.

REPLY: We have cleaned the manuscript for typos.

All in all though this is a thoughtful and interesting contribution that is definitely worth publishing once these minor concerns have been addressed.

Reviewer #3

This is a very interesting paper with, I think, plausible models for the interactions of modern humans and Neanderthals across contact zones, and the potential roles of pathogens on those interactions.

7. COMMENT: I have marked up some small points on the main manuscript (although I left American spellings for words like behaviour and Palaeolithic)

REPLY: We have accepted and corrected the small points mentioned.

One comment was “modeled or modelled?”. We use the US spelling “modeled”, throughout the manuscript.

Another comment was “Ust-Ishim suggests hybridization had already occurred ~55 ka?”. We address this issue in comment 9 below.

... and I have no competence to comment on the modelling methods and results. My comments centre on the general approach and the context of the questions that are addressed. I also note below some other literature that might be relevant here.

8. COMMENT: My main point is that this paper assumes that there was a long-lasting and localised inter-species boundary between moderns and Neanderthals in the Levant. There may well have been, but it is by no means certain that this boundary occurred uniquely in the Levant rather than, say, Arabia or Iraq, nor is there yet adequate chronological control to even show that these populations actually overlapped in the Levant at any one time, although it is likely they did, of course. These limitations/qualifications need to be admitted near the beginning of the paper, before proceeding to the assumptions that are used for the modelling.

REPLY: Yes, this point was also made by reviewer 2. We have added the point suggested by the reviewers in lines 91-93, and we address the modeling considerations in lines 398-403.

9. COMMENT: Another assumption seems to be that moderns did not extend beyond the Levant until destabilisation of the interaction front around 45-50 ka. However, there is already a modern human fossil at Ust-Ishim in Siberia directly dated at about 45 ka, with an estimated introgression date for its Neanderthal DNA of ~55 ka. And as discussed in, for example, Bae et al. *Science* 358, eaai9067 (2017), there is growing evidence that moderns extended beyond the Levant well before 50 ka, and probably before 80 ka, so how does that affect the arguments of this paper about disease constraints on early moderns in Eurasia? Indeed, there is a paper in press that seemingly takes *H. sapiens* beyond the Levant at an even earlier date. Additionally, genetic arguments have been advanced for an earlier interaction of the sapiens and Neanderthal lineages, as reflected by a probable Middle Pleistocene spread of mtDNA (Posth et al. *Nature Communications* volume 8, Article number: 16046 2017). Even if the authors question some of these findings or somehow consider them irrelevant to their analyses, I think they do need to at least discuss the data that suggest more complex scenarios than those that they model.

REPLY: Accepted. We have added these references and addressed the applicability of our models to more complex scenarios in lines 398-403. To address some of the specific evidence mentioned by the reviewer:

(1) We believe that the Ust Ishim findings are consistent with our description of the timeline. The Ust-Ishim fossil belongs to a modern human, estimated to have lived 45kya (Fu et al., 2014). Genetic evidence suggest that the ancestors of this individual interbred with Neanderthals 7-13kya before he lived (Fu et al., 2014). Therefore, the finding of a modern human fossil in Siberia 45kya with an introgression signature ~10k years earlier is consistent with a modern human expansion across Eurasia at 45-50kya, and a hybridization event of its ancestors occurring before 50kya, possibly in the Levant.

(2) The mtDNA evidence for early gene flow in Potsh et al., before 100kya but later than 270kya, is consistent with the introgression in the contact zone that we describe, since modern humans have been found in the Levant at least as early as 130kya, and perhaps even earlier. We have added this reference as supporting evidence in line 73.

10. COMMENT: Finally, I was interested enough by this paper to investigate whether there is already a relevant literature for non-human species, and indeed there is. For example (in this case concentrating on parasites) Theodosopoulos et al. Parasites and Host Species Barriers in Animal Hybrid Zones 2018DOI:<https://doi.org/10.1016/j.tree.2018.09.011>

Species barriers are tested in hybrid zones when gene flow occurs between hybridizing species. Hybridization can erode species barriers, lead to the introgression of adaptive traits, or remain stable through time. Outcomes in hybrid zones are influenced by divergence between the hybridizing taxa, behavior, ecology, and geography. Parasites and pathogens play a major role in host fitness and appear to have varied impacts on species barriers in hybrid zones. We comprehensively reviewed the literature on parasitism in animal hybrid zones and present an evolutionary framework within which to consider parasite–hybrid interactions. Parasites most frequently show potential to contribute to species barrier breakdown in hybrid zones, but also frequently show potential to facilitate the maintenance of species barriers. Incorporating eco-immunology, parasite community theory, and spatiotemporal approaches will be important as genomic tools allow researchers to examine parasites and hybrid zones at greater resolution and in a diversity of natural habitats.

REPLY: Thank you for bringing to our attention this interesting review paper. A similar point was made by reviewer 1. We now address the similarity between the dynamics we describe and animal examples, including the reference mentioned by the reviewer, in lines 355-364.

Reviewer #4

Greenbaum et al present a modelling study of interactions between modern humans and Neanderthals in the Levant, in the period before the out of Africa episode of modern humans that occurred 45 to 50 thousand years ago. Their study aims to explain the long period of stable coexistence of the two species in the Levant before the sudden modern humans spread toward Eurasia. The basic hypothesis is that each of the two species would have carried a package of pathogens from the environment in which they evolved (temperate for Neanderthals and tropical for modern humans), to which the other species would have been vulnerable. The models developed show that this hypothetical configuration well explains a long period of stable coexistence between the two species and that introgression of resistant genes due to admixture between them would explain the gradual acquisition of resistance in each of the two species. Models show that if demographic and disease conditions are symmetrical, then neither species takes an advantage over another. However, if conditions are asymmetric, then one species can get rid of its disease burden faster than the other, and thus expand over the other species range. According to the results presented, an asymmetry in the pathogen packages could theoretically explain both the long period of contact between modern humans and Neanderthals in the Levant and the sudden expansion of modern humans to the detriment of Neanderthals.

I find the paper very well written, clear and the proposed hypothesis interesting and well supported by the literature cited. The proposed models seem robust, but I was not able to verify all its mathematical background.

I am thus in favor of publishing this article once the points below have been addressed.

My main concerns regard an aspect that I do not think sufficiently addressed in the paper: the densities of interacting populations.

11. COMMENT: First, the authors hypothesize that epidemics in Neanderthals and modern humans could explain the long period of contact between them in the Levant. However, were the densities of populations of these two species and the connectivity between them in the area sufficient to allow epidemics to spread? The densities of these two species are usually thought to be very low at this time (eg Bocquet-Appel et al, *Antiquity* 2000, *J Archaeol Sci* 2000). Would such low densities be in accordance with the model's assumptions?

REPLY: We agree with the reviewer that in order to understand the frequency and characteristics of epidemics, we would need to know more about population densities, population structure, and also behavior, of both species. The genomic evidence, such as the genomic signature of positive selection on introgressed immune-related genes during this time period, supports the idea that epidemics did indeed spread, and had a strong enough impact on survival and fitness that the selection signal is still detectable today (lines 73-80 and 365-370).

12. COMMENT: Secondly, the results of the study show that the species with the lowest density will get rid of its disease burden the fastest and thus be able to spread further. However, it is generally assumed that modern human densities was higher than density of Neanderthals (eg

Currat et al, PNAS 2011 and references above), which would contradict the results presented. We can also think that modern humans living in Africa constituted a reservoir for modern humans in the Levant, so their population would be larger than the Neanderthal population. Do the authors have arguments in favor of a lower density of modern humans than Neanderthals in the Levant?

REPLY: Our result shows that for most of the parameters we examined, the species with lower initial density (i.e. at the time of initial contact) would overcome disease burden sooner (lines 254-258). While it is likely that modern human density was higher than Neanderthals' in general, it may have been that at the initial encounter between the species, as modern humans were beginning to migrate out of Africa, modern human densities were lower than Neanderthals. However, we have no direct evidence for this; we therefore suggest that more detailed examination of relative densities over time may shed light on some of our predictions (lines 389-397).

13. COMMENT: In addition, in the second paragraph of page 9, there is a contradiction between the results of the first and second parts. In the first part, species 2 suffers from higher mortality (and therefore presumably lower density), but species 1 gets rid of its disease burden the fastest. This last result is in contrast to part 2 below where the less dense species 1 is the first to recover from burden.

REPLY: It seems we have not been clear enough in this section regarding the scenarios we discuss. In the first part of this paragraph (lines 249-254), species 2 suffers higher mortality due to a more substantial pathogen package of species 1, and the densities of species 1 and 2 are assumed to be the same. In this case, we find that species 1 recovers sooner from disease burden. In the second part of this paragraph (lines 254-258), we assume the species carry similar pathogen packages, but species 1 is less dense than species 2. In this case we find that the less dense species, species 1, is first to remove its disease burden. We have now stated the differences between the scenarios we examined more clearly, in lines 254-258.

14. COMMENT: Third, according to the model proposed by the authors, the expansion of humans in Eurasia would have occurred from the Levant, and not from Africa as supported by many studies (eg Prugnolle et al, Current Biology 2005). How can the authors explain this discrepancy?

REPLY: The dynamics of expansion that we are attempting to clarify in the manuscript concern expansion of modern humans from Africa to the Levant, as early as 130kya, and then further expansion into Eurasia at around 45-50kya. Modern humans certainly expanded into Eurasia from Africa, observing the expansion at a broad enough time-scale. However, there is a likely possibility that modern humans first expanded into the Levant from Africa, and later expanded deeper into Eurasia from there. Therefore, there is no contradiction between the scenarios we examine in the manuscript and the genetic evidence of an African origin to modern humans,

such as in Prugnolle *et al.* 2005 (and the similar work of some of our team, Ramachandran et al. 2005).

I also noted a few edits:

15. COMMENT: Page 2, end of the fourth paragraph: “suggest suggests”, one “suggest” should be removed.

REPLY: Corrected (line 80).

16. COMMENT: Page 8, fifth paragraph: “single-time-scale model model”, one “model” should be removed.

REPLY: Corrected (line 231).

17. COMMENT: Page 9, last paragraph: “taking by force, individuals of the other species”, it is probably meant “population” instead of “species” in that context.

REPLY: We clarified our point here (line 284), which is that inter-species contact rates could be modified through behavior. Changed to “...taking by force, individuals from bands of the other species.”

Reviewers' Comments:

Reviewer #4:

Remarks to the Author:

I am satisfied with the answers to my comments.